# Effects of Atmospheric Ammonia on Skeletal Muscle Growth in Broilers

**DOI:** 10.3390/ani13121926

**Published:** 2023-06-09

**Authors:** Xin Zhao, Guangju Wang, Hongyu Han, Ying Zhou, Jinghai Feng, Minhong Zhang

**Affiliations:** State Key Laboratory of Animals Nutrition, Institute of Animal Sciences, Chinese Academy of Agricultural Sciences, Beijing 100193, China; zhaoxin09@caas.cn (X.Z.); guangju.wang@wur.nl (G.W.); hhycaas@163.com (H.H.); zhouying@caas.cn (Y.Z.); fjh6289@126.com (J.F.)

**Keywords:** atmospheric ammonia, skeletal muscle growth, muscle fibre type, insulin-like growth factor 1, myostatin pathway

## Abstract

**Simple Summary:**

Ammonia is the most prominent irritant gas emitted by poultry houses, mainly from animal wastes produced by microbial activity. If the atmospheric ammonia concentration cannot be controlled in a low range, it will seriously affect poultry production and health. In recent years, more and more attention has been paid to the adverse effects of atmospheric ammonia concentration on poultry. However, there are few studies on the effect of ammonia on skeletal muscle growth. This study aims to explore the effect of ammonia on skeletal muscle growth in broilers.

**Abstract:**

Ammonia, one of the most polluted gases in poultry houses, has always been an urgent problem to solve. Exposure to ammonia can threaten the respiratory tract, induce inflammation, and decrease growth performance. To date, there are few studies investigating the effects of ammonia on skeletal muscle growth. In this experiment, a total of 144 broilers were randomly divided into two groups, and 0 ppm and 35 ppm atmospheric ammonia were administered in the chambers. The trial lasted for 21 days. The breast muscle, thigh muscle, dressed weight, and serum biochemical indexes were measured. The skeletal muscle fibre morphology was observed using light microscopy, and the expressions of genes associated with skeletal muscle development and myosin heavy chain genes were assessed. After 7 days of ammonia exposure, the broilers’ weight in the ammonia group decreased. On the 21st day of the experiment, in the ammonia group, the breast muscle weight, thigh muscle weight, and dressed weight decreased, the blood urea nitrogen content increased, skeletal muscle fibre diameter shortened, the expression of myostatin increased, and the expression of myosin heavy chain-FWM and myosin heavy chain-FRM decreased significantly. This article suggests that 35 ppm atmospheric ammonia seriously affects the skeletal muscle gain rate of broilers, and the myostatin pathway could be a potential regulation of the growth rate of muscle fibre under ammonia exposure.

## 1. Introduction

Atmospheric ammonia (NH_3_), one of the most important hazardous gas in poultry houses [1], has long been recognized as a significant environmental problem in poultry grow-out facilities negatively affecting poultry performance. Birds are more sensitive to ammonia than other animals [2]. Previous studies have demonstrated that dressed weight, weight gain, dressing percentage, and thigh and breast muscle percentage were reduced when ammonia is above 25 ppm [3,4]. Since breast muscle and thigh muscle of broilers are the main part of chicken meat production. The adverse effects of ammonia on skeletal muscle growth cannot be ignored. Therefore, it is very necessary to explore the effect of atmospheric ammonia in the poultry house on the growth of broiler skeletal muscle and its mechanism. However, so far, atmospheric ammonia’s effects on skeletal muscle growth and its mechanism in broilers remain unclear.

The growth phase of broilers is a crucial period of skeletal muscle development because the muscle fibre numbers are constant after birth [5]. Skeletal muscle hypertrophy through the enlargement of individual muscle fibres is observed during the growth period. Myonucleus is nondivision in muscle fibre. Satellite cells are the main source of new nuclei during skeletal muscle growth or regeneration. There is wide research on the effect of growth factors and hormones on the proliferation and differentiation of muscle cells during muscle growth and regeneration. The growth of skeletal muscle in avians is regulated by many factors. For instance, typically, insulin-like growth factor 1 (IGF-1) and myostatin (MSTN) play critical roles in the growth of skeletal muscle [6]. The IGF-1 shows a positive role in regulating skeletal muscle growth. IGF-1 is known to activate the PI3K-Akt pathway, which can induce hypertrophy of transfected muscle fibres [7]. A study has indicated that IGF-1 can stimulate the proliferation and differentiation of myoblasts [8]. In addition, several studies have shown that the IGF-1 signalling pathway promotes muscle growth and inhibits protein decomposition [9,10,11]. Conversely, the MSTN pathway plays a negative regulatory role, i.e., inhibiting satellite cell proliferation and differentiation and promoting protein degradation. The purified MSTN inhibited protein synthesis and reduced the size of the myotube [12]. Moreover, muscle atrophy occurred in healthy mice after the administration of MSTN [13].

Skeletal muscle exists in the form of a collection of substantial muscle fibres. Based on the composition of myosin heavy chain (MyHC), skeletal muscle fibres can be classified into I, IIa, and IIb types in avian [14,15]. The skeletal muscle fibre exhibits remarkable plasticity, of which fibre type switching is an example. Exercise, environment, and stress could induce a mutual switch between skeletal muscle fibres [16]. However, there have been few studies on the effect of ammonia exposure on muscle fibre types in broilers. In addition, a previous study showed that the proportion of fast muscle and glycolytic phenotype increased in fetal mice with MSTN knockout [17], suggesting that the MSTN pathway could regulate skeletal muscle growth through muscle fibre type transformation during animal skeletal muscle development. Thus, it is of special importance to understand the connections between the MSTN pathway and muscle fibre types in broilers. However, there is no report on the effect of ammonia on gene expression associated with skeletal muscle and fibre types in broilers so far.

Probing the mechanism of slow skeletal muscle growth under stress contributes to breeders formulating countermeasures. Therefore, the present study explores the effects of atmospheric ammonia on skeletal muscle mass, muscle fibre types, and genes associated with muscle growth in broilers. Furthermore, this study intends to provide a preliminary investigation of the mechanism of skeletal muscle growth.

## 2. Materials and Methods

### 2.1. Experimental Design

A total of 164 0-day-old male Arbor Acres (AA) broilers were purchased from the hatchery and the experiment started after 21 days of feeding. A total of 144 birds of similar weight were selected and randomly divided into two groups with six replicates in each group (12 birds for each replicate), and the birds were housed in the experimental chambers. Temperature and humidity were maintained at 21 ± 1 °C and 60 ± 7%, respectively. The concentrated NH_3_ was delivered to chambers with different concentrations of 0 ± 3 ppm and 35 ± 3 ppm, respectively. The concentrations of NH_3_ in the chambers were monitored with a LumaSense Photoacoustic Field Gas-Monitor INNOVA 1412 (Santa Clara, CA, USA) during the entire experiment. The basal diet composition was presented in Table 1 and Table 2. Diet and drinking were supplied ad libitum. The experiment was approved by the animal welfare and ethics checklist of the Institute of Animal Science, Chinese Academy of Agricultural Sciences (permit number: IAS 2021-75).

### 2.2. Sample Collection

At the end of each test period (7 d and 21 d), the cervical dislocation was used for euthanized in birds of the control group and the ammonia group (one bird each replicate). After anesthesia, the blood samples were collected, the half breast muscle and thigh muscle were removed, and then the half weight of the breast muscle and thigh muscle were recorded. Blood serum, liver tissue, and breast muscle were obtained and stored at −80 °C for further analyses. The part of the breast muscle tissue was taken, soaking in the paraformaldehyde for the paraffin sections made.

### 2.3. Serum Biochemical Indexes Analysis

At 7 and 21 days, the serum levels of total protein (TP), albumin (ALB), and blood urea nitrogen (BUN) were measured using testing kits (Nanjing jiancheng Bioengineering Institute, Nanjing, China).

### 2.4. Morphology and Analysis of Skeletal Muscle Tissue

Samples were cut from breast and thigh muscle tissue for fiber diameter measurement. Cured at 4 °C for 24 h, soaked in 20% nitric acid for 24 h, added 70% glycerol to the glass slide. Use a dissecting needle to evenly disperse the muscle fibers, then place a coverslip on the samples. The muscle fiber diameter was measured using an eyepiece micrometer (BX41, Olympus (China) Co., Ltd., Shanghai, China). Take the paraformaldehyde-fixed tissue block, trim it to a suitable size, and after dehydration and transparency, carry out routine paraffin embedding, and a 4 μm section. After the sections were deparaffinized with xylene and washed with various levels of ethanol, stained with hematoxylin for 5 min, differentiated with hydrochloric acid and ethanol for 30 s, immersed in distilled water for 15 min, and then stained with 1% eosin solution for 2 min. After normal dehydration, seal it transparently. Select the field of view for each slice in each group to take pictures, and use Image-Pro Plus 6.0 software to measure the diameter and number of muscle cells based on the 200-fold ruler.

### 2.5. RNA Extraction and Reverse-Transcription PCR

Breast muscle and thigh muscle specimens from randomly selected broilers were ground separately in a liquid nitrogen frozen state. The general total RNA extraction kit (Hooseen biology, Beijing, China) was used for sample RNA extraction from different muscle tissues. Using Nano-100 and denaturing agarose gel electrophoresis to detect RNA concentration, purity, and degradation degree, which confirms that the quality of the isolated RNA is acceptable and can be used throughout the rest of the protocol. Use Takara Reverse Transcription Kit for qualified samples, to avoid the influence of DNA; an amount of 1 µg of total RNA was incubated with DNase I (1 U/µg RNA; Invitrogen, Waltham, MA, USA), followed by inactivation of DNase I by adding EDTA and heating at 65 °C for 5 min. RNA was then reverse transcribed using SuperScript III (200 U/µL; Invitrogen) and oligo-d(T) primers (Invitrogen). Reactions were performed at 42 °C for 15 min, at 50 °C for 50 min, and at 70 °C for 15 min to inactivate the enzyme. SYBR Premix Ex Taq II was used for RT-PCR processing. The primers were synthesized based on predicted chicken sequences by Sangon Biotech (Shanghai, China) in Table 3. Reactions were performed at 42 °C for 15 min, at 50 °C for 50 min, and at 70 °C for 15 min to inactivate the enzyme, and store cDNA in a refrigerator at −20 °C for later use.

### 2.6. Relative Quantitative Real-Time PCR Analysis

In order to measure the relative quantitative expression of genes (MSTN, IGF-1, MyHC-FWM, MyHC-FRM, and MyHC-SM) in skeletal muscle samples of the 35 ppm group and the control group, relative quantitative real-time PCR was performed using an SYBR^®^ Premix Ex Taq™ II (Tli RNaseH Plus) kit (Takara Bio Inc., Kusatsu, Japan). The PCR program was performed on the ABI 7500 Real-Time PCR Detection System (Applied Biosystems, Foster City, CA, USA) using the SYBR Premix Ex Taq II Kit (Takara Bio) at 95 °C for 15 s and 60 °C 40 cycles of 30 s. All measurements were performed in triplicate. Fold differences were calculated using the ΔΔ Ct method, and data were normalized using the geometric mean of 18S rRNA and glyceraldehyde 3-phosphate dehydrogenase (GAPDH) mRNA (Δ Ct). The relative gene expression amount of each sample was calculated according to the formula for relative quantities: RQ = 2^−ΔΔ Ct^.

### 2.7. Data Analysis

The SPSS 19.0 statistics software was used to analyze numerical data. The independent sample *t*-test was used to compare the values of the two groups. The results data were presented as mean ± SEM (standard error of the means). *p* < 0.05 was regarded as a significant difference.

## 3. Results

### 3.1. Effect of Ammonia Exposure on the Muscle Weight

The effect of 35 ppm ammonia exposure on muscle weight is presented in Table 4. On the 7th day, there is no significant difference observed in breast and thigh muscle weight between the 35 ppm group and the control group, while the body weight of the 35 ppm group was significantly reduced (*p* < 0.001) compared with the control group. It is worth noting that the *p* value of breast muscle weight and thigh muscle weight difference in the two groups is 0.1013 and 0.1068, respectively, which means after 7 days of ammonia exposure, the muscle weight gradually tends to be significantly different. On the 21st day, the breast muscle weight (*p* < 0.001), thigh muscle weight (*p* < 0.001), and body weight (*p* < 0.001) of the 35 ppm group were significantly reduced compared with the control group.

### 3.2. Effect of Ammonia Exposure on Serum Biochemical Indexes

We used ELISA to measure the content of serum biochemical indexes. The effect of 35 ppm ammonia on the serum biochemical indexes of broilers is presented in Table 5. On the 7th day, there was no difference in the level of serum total protein (TP), albumin (ALB), and blood urea nitrogen (BUN) in broiler chickens between the 35 ppm group and the control group. However, it is clear that the level of BUN increased to some extent under 35 ppm ammonia exposure, even though there is no statistical significance. On the 21st day, the BUN level in the 35 ppm group increased significantly compared with the control group (*p* < 0.01). However, there was no significant difference in TP and ALB between the two groups.

### 3.3. Effect of Ammonia Exposure on Skeletal Muscle Fibres and Tissue Morphology

To further analyze the response of broiler skeletal muscle to exogenous ammonia gas, we performed hematoxylin-eosin staining of the breast skeletal muscles of broilers that were exposed to 35 ppm ammonia concentration for 21 days and the control group. The statistical analysis results of the properties of skeletal muscle fibres are shown in Table 6. The results of hematoxylin-eosin staining skeletal muscle tissue e are shown in Figure 1. After being exposed to 35 ppm ammonia for 21 days, the skeletal muscle cell diameter decreased significantly (*p* < 0.05), while the skeletal muscle fibre density increased significantly (*p* < 0.05) compared with the control group. After being exposed to 35 ppm ammonia for 21 days, according to the results shown in Figure 1, it can be clearly seen that the muscle fibers of the control group are thicker than those of the 35 ppm group. In addition, the presence of gaps can be clearly seen in the control group, while in the 35 ppm group, it is almost invisible due to the high density. Since the number of muscle fibers is constant after birth, the diameter of the muscle bundles is also shortened due to the general shortening of muscle fiber diameters. Eventually, it results in lower weight for the muscle tissue.

### 3.4. Effect of Ammonia Exposure on the Expression of Genes Associated with Skeletal Muscle Growth

To further analyze the effect of exogenous ammonia exposure on the growth of skeletal muscle, we analyzed the mRNA expression of genes associated with skeletal muscle growth in the skeletal muscle of the control group and the 35 ppm group after 21 days of ammonia exposure. The qPCR results are shown in Figure 2. After being exposed to 35 ppm ammonia for 21 days, the mRNA expression of the MSTN gene in skeletal muscle was significantly up-regulated (*p* < 0.05), and the expression of IGF-1 had a downward-regulated trend (*p* < 0.1). Moreover, the expression of MyHC-FWM and MyHC-FRM genes in the skeletal muscle of the 35 ppm group was significantly decreased (*p* < 0.05), whereas the mRNA expression of the MyHC-SM had no statistical difference between the two groups.

## 4. Discussion

Ammonia is one of the most important polluting gases in poultry houses and has adverse effects on the health of broilers. Since commercial broilers such as Arber Acres were intensively selected in the broiler breeding program, the resilience is relatively low compared with other normal breeds. Breast and thigh muscle weights are important indexes in broiler breeding. Previous studies have shown that exposure to ammonia seriously threatens the broilers’ health, which not only leads to respiratory disease [18] but also slows down the growth of skeletal muscle and reduces the growth rate of breast, thigh muscle, and body weight [3]. More seriously, exposure to ammonia could increase the morbidity of broilers [19], resulting in low breeding efficiency. On the 21st day of this trial, broilers’ growth performance in the 35 ppm group was impaired under the ammonia treatment. The average breast, thigh muscle, and body weight of the control group were higher than that of the 35 ppm group, which indicated the growth of skeletal muscle and performance were reduced after 21 d ammonia exposure. Interestingly, there was no significant difference between the two groups in breast muscle and thigh muscle weight on day 7. The values observed in the 35 ppm group remained less than that of the control group, indicating that the degree of injury increased with the prolongation of exposure time under 35 ppm ammonia exposure.

The change in skeletal muscle mass is determined by the balance between protein synthesis and protein degradation [20]. In general, skeletal muscle hypertrophy occurs when the protein synthesis rate is higher than the degradation rate during skeletal muscle growth [21]. In adulthood, relatively, stable skeletal muscle weight is due to balanced muscle protein synthesis and degradation [22]. In broiler breeding, Arbor Acres broilers grow fast, and the protein synthesis rate is much greater than the degradation rate in their growth stage. Urea nitrogen is the main end product of protein degradation. TP, ALB, and BUN can be used as markers of protein metabolism [23]. In the current study, after 21 days of ammonia exposure, the BUN of the ammonia group increased significantly. We have mentioned that BUN is the final product of protein degradation. Hence, the increase in BUN level indicates an increase in protein degradation rate, in turn, produce more BUN in protein metabolism, which is consistent with the ammonia group’s decrease in skeletal muscle weight.

The body weight and breast and thigh muscle weight of broilers are related to the size of skeletal muscle fibers, and the diameter and density of muscle fibres. Muscle fibre hypertrophy (increased diameter and area of muscle fibres) occurs during the skeletal muscle growth phase mainly by adding proteins and nuclei derived from satellite cell proliferation and fusion [24]. In the current study, the density of muscle fibres increased. However, the diameter of muscle fibres decreased in the ammonia group. While the number of muscle fibres increases only during the embryonic period, muscle fibres remain constant after hatching [24]. Therefore, This means that the volume of skeletal muscle is smaller compared to broiler chickens not exposed to 35 ppm ammonia. Furthermore, it shows that the degradation of protein in muscle fibres intensified, slowing down the deposition rate of protein, eventually leading to a slow weight gain of skeletal muscle.

IGF-1 can regulate the synthesis and degradation of proteins, and its main function is to promote protein synthesis. The expression of IGF-1 in skeletal muscle is related to the hypertrophy and function of muscle fibres [25]. Protein synthesis is stimulated by IGF-1 by upregulating PI3K, Akt, and mTOR phosphorylation [26]. The overexpression of IGF1 in mouse muscles caused a two-fold increase in muscle size by suppressing protein breakdown and atrophy-related ubiquitin ligases, MAFbx, and MuRF1 expression [10]. A previous study has shown that ammonia exposure can reduce the expression of the IGF-1 gene in pufferfish [27]. Thus, the decrease in IGF-1 could be a reason for protein synthesis slowing down. In the current study, we observed that the IGF-1 content of the ammonia group has a downward trend, but it did not show significance, which may be the result of an insufficient ammonia exposure period or insufficient ammonia concentration. As a negative regulator of skeletal muscle growth, overexpression of MSTN can lead to significant atrophy of skeletal muscle [28]. The proteolysis of skeletal muscle was mainly regulated by the MSTN pathway. MSTN was considered a core factor in the MSTN pathway [6]. Indeed, when either of the two pathways mentioned above is activated, it will inhibit the activity of the other [29]. Therefore, the increased MSTN expression indicated that the MSTN pathway was activated, and the skeletal muscle protein degradation rate increased while inhibiting the activity of the IGF-1 pathway, leading to limited protein synthesis.

Different skeletal muscle fibre types have different biochemical properties and metabolic characteristics [30]. There are many approaches to classifying skeletal muscle fibres, and the most commonly used one is the MyHC isoform. Muscle fibre type can be affected by environmental factors and physiological state and change the expression of muscle fibre-specific genes through intracellular related signal pathways, thus triggering the transformation of muscle fibre type [31]. The relative percentages of specific muscle fibre types can be estimated by quantifying the expression level of the MyHC gene in skeletal muscle [32]. In the current study, the expression of MyHC-FRM and MyHC-FWM were significantly reduced, both of which were fast muscles and had lower oxidation ability. Compared with the fibres with high oxidation ability, the muscle fibres with low oxidation ability are relatively larger [33]. Therefore, under the influence of ammonia, there seems to be a synergistic effect between the change in muscle fibre type and the size of muscle fibre, indicating that ammonia could affect muscle growth by triggering the transformation of muscle fibre type.

Recent evidence suggested that skeletal muscle fibre type switching is related to the expression of MSTN [34]. Furthermore, the previous report has shown that mice with MSTN knockout show a higher proportion of fast muscle and glycolysis MyHC isoforms [17]. This may indicate that the MSTN pathway may be involved in the induction of muscle fiber transformation by various factors. In the current study, the expression of MSTN increased, while the proportion of fast-twitch muscle fibres decreased significantly in broilers exposed to ammonia, indicating the decrease in the proportion of fast muscle in skeletal muscle after ammonia exposure may be related to the activation of the MSTN pathway. Moreover, regarding how the MSTN pathway is activated by ammonia and the changes in the pathway, future studies should focus on these issues.

## 5. Conclusions

In conclusion, after 21 days of ammonia exposure, the expression of MSTN was increased, the muscle protein synthesis rate reduced, the degradation rate increased, and the diameter of the muscle fibres tapered. The proportion of fast-twitch muscle decreased, leading to a decrease in skeletal muscle protein deposition rate and slow growth of skeletal muscle weight and body weight in broilers. Our results show that atmospheric ammonia can slow down the muscle growth rate, and the myostatin pathway could be a potential regulation of the growth rate and the transformation of muscle fibres under ammonia exposure.

## Figures and Tables

**Figure 1 animals-13-01926-f001:**
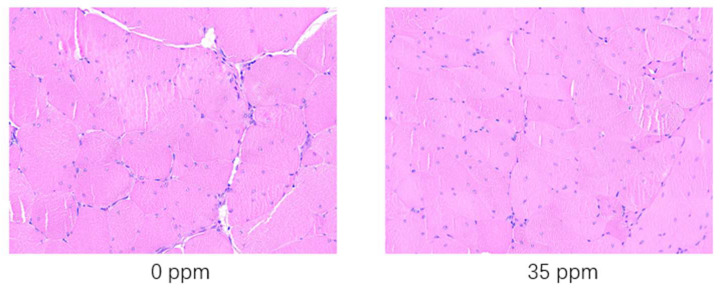
HE staining (scale bar = 50 μm) results of the skeletal muscle of broiler chickens in the ammonia group and control group.

**Figure 2 animals-13-01926-f002:**
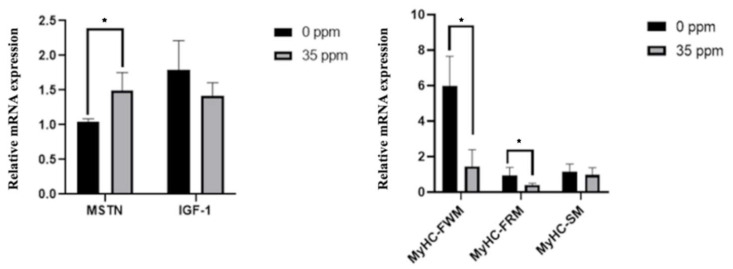
The effect of ammonia on the expression of genes associated with skeletal muscle growth (21 days). * Indicate the difference is significant at 0.05 level.

**Table 1 animals-13-01926-t001:** Ingredients compositions of the basal diet.

Ingredients (g/kg)	Content (%)
Corn	56.51
Soybean meal	35.52
Soybean oil	4.50
NaCl	0.30
Limestone	1.00
Dicalcium phosphate	1.78
DL-Methionine	0.11
Premix ^1^	0.28
Total	100.00

^1^ Premix provided the following per kg of the diet: vitamin A, 10,000 IU; vitamin D3, 3400 IU; vitamin E, 16 IU; vitamin K3, 2.0 mg; vitamin B1, 2.0 mg; vitamin B2, 6.4 mg; vitamin B6, 2.0 mg; vitamin B12, 0.012 mg; pantothenic acid calcium, 10 mg; nicotinic acid, 26 mg; folic acid, 1 mg; biotin, 0.1 mg; choline, 500 mg; Zn (ZnSO_4_·7H_2_O), 40 mg; Fe (FeSO_4_·7H_2_O), 80 mg; Cu (CuSO_4_·5H_2_O), 8 mg; Mn (MnSO_4_·H_2_O), 80 mg; I (KI) 0.35 mg; Se (Na_2_SeO_3_), 0.15 mg.

**Table 2 animals-13-01926-t002:** Nutrient compositions of the basal diet.

Calculated Nutrient Levels	Content
Metabolizable energy (MJ/kg)	12.73
Crude protein (g/kg)	20.07
Available Phosphorus (g/kg)	0.40
Calcium (g/kg)	0.90
Lysine (g/kg)	1.00
Methionine (g/kg)	0.42
Methionine + cysteine (g/kg)	0.78

**Table 3 animals-13-01926-t003:** Sequences of primers used in PCR experiments.

Gene Name	Gene ID	Primer Sequence
MyHC-FWM	768566	GTCTGGAGAAGACATGCCGA
		ACGAGCTCTTTGAGCATTAACATC
MyHC-FRM	417310	ATTCACGGCAGGTGGAAGAA
		TCCTCTGTGCGCTGAATAGC
MyHC-SM	U85022	CGAGGAGAAGGCCAAGAAGG
		TTCTTCATCCGCTCCAGGTG
IGF-1	418090	GAGTTGTGACCTGAGGAGGC
		TTTGGCATATCAGTGTGGCG
MSTN	373964	TGCAATGCTTGTACGTGGAG
		AGTGGAGGAGCTTTGGGTAAA
Reference gene	425619	TTGCTGCTGGAGATGACAAG
		CTTCTTGATACGCCTGTTG

**Table 4 animals-13-01926-t004:** Effect of ammonia on skeletal muscle weight.

		Breast Muscle Weight/g (Half)	Thigh Muscle Weight/g (Half)	Body Weight/kg
Day 7	0 ppm	111.68 ± 4.07	94.26 ± 3.75	1.31 ± 0.03 ^a^
	35 ppm	100.18 ± 5.22	86.39 ± 2.68	1.15 ± 0.02 ^b^
		0.1013	0.1068	*p* < 0.05
Day 21	0 ppm	318.43 ± 10.28 ^a^	233.68 ± 6.58 ^a^	3.05 ± 0.02 ^a^
	35 ppm	246.63 ± 8.35 ^b^	183.97 ± 5.16 ^b^	2.45 ± 0.05 ^b^
		*p* < 0.05	*p* < 0.05	*p* < 0.05

Values are means ± SEM. ^a, b^ Means within the same line with different superscripts differ significantly (*p* < 0.05).

**Table 5 animals-13-01926-t005:** Effect of 35 ppm ammonia on serum biochemical indexes.

		TP g/L	ALB g/L	BUN mmol/L
Day 7	0 ppm	26.57 ± 0.46	12.07 ± 0.24	0.25 ± 0.06
	35 ppm	27.07 ± 0.64	12.37 ± 0.41	0.41 ± 0.07
		-	-	-
Day 21	0 ppm	28.75 ± 1.21	12.97 ± 0.38	0.28 ± 0.03 ^b^
	35 ppm	28.5 ± 1.07	11.67 ± 0.63	0.49 ± 0.05 ^a^
		-	-	*p* < 0.05

Values are means ± SEM. TP, total protein; ALB, albumin; BUN, blood urea nitrogen. ^a, b^ Means within the same line with different superscripts differ significantly (*p* < 0.05).

**Table 6 animals-13-01926-t006:** Effects of ammonia exposure on the properties of skeletal muscle fibres (21 days).

	Diameter of Muscle Cells/mm	Muscle Fibre Density n/mm^2^
0 ppm	0.090 ± 0.007 ^a^	139.35 ± 8.68 ^b^
35 ppm	0.075 ± 0.008 ^b^	205.57 ± 14.33 ^a^
	*p* < 0.05	*p* < 0.05

Values are means ± SEM. TP, total protein; ALB, albumin; BUN, blood urea nitrogen. ^a, b^ Means within the same line with different superscripts differ significantly (*p* < 0.05).

## Data Availability

Not applicable.

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
