# Peer review of "Effects of Atmospheric Ammonia on Skeletal Muscle Growth in Broilers"

_animals, 2023, doi:10.3390/ani13121926_

Round 1

Reviewer 1 Report

Comments:

I’ve the following concerns.

1.           Check the English carefully. Check the sentence structure, spelling, verbs, and remove extra spaces, and add where needed.

2.           Write the full form of all the abbreviations used in this manuscript.

3.           What is the source of ammonia and how its concentration is maintained to 35ppm? 

Author Response

Dear reviewer,

We gratefully appreciate for your valuable comments and suggestions.

We have carefully checked the mistake and revised the manuscript according to your suggestions.

Considering your main concerning, we used a LumaSense Photoacoustic Field Gas-Monitor INNOVA 1412 (Santa Clara, CA, USA) to monitored the concentrations of NH3 in the chambers which maintain the concentration of NH3 at 35 ppm during the entire experiment.

Kind regards,

Xin Zhao, Guangju Wang

Reviewer 2 Report

Dear Editor,

the manuscript by Zhao et al. interestingly describes the effects of atmospheric ammonia on muscle growth in poultry. The article is indeed well written, I only have some minor corrections.

-Line 5: would be better to use institutional emails.

-Line 15-29: please delete headings. See authors' instructions.

- Line 78: can you add a picture of poultry in the controlled chambers? would be interesting for the readers.

-How ammonia was added in the cambers and controlled ? Please, give more details. 

- The temperature- and humidity-controlled chambers needs to be explained and showed.

-Lines 83-84: please use capital letters "Institute of Animal Science, Chinese Academy of Agricultural Sciences.

- Table 1: could be bettere to see two tables. I mean one for ingredients and one for nutritional values. There were only corn and soybean as "feed" ?

Author Response

Dear reviewer,

Thanks for your helpful comments and suggestions, we have carefully checked the mistakes and revised the manuscript according to your suggestions.

In the manuscript, we have replaced part of personal emails with institutional emails, please see line 5.

We have deleted headings in the abstract, please see lines 15-29.

Considering the confidentiality of climate chambers in our institute, we are not allowed to share the photos on the publication, so we are sorry that we cannot upload the photos of broilers in chambers.

We used a LumaSense Photoacoustic Field Gas-Monitor INNOVA 1412 (Santa Clara, CA, USA) to monitor the concentration of NH3 during the entire experiment, and we added this information to the manuscript.

The chambers we used in the experiment can also control the temperature and humidity.

We have adjusted the sentences in the experimental design part, please see line 101.  

We have corrected the spelling of the institute name, please see line 108.  

We have divided Table 1 into two tables, please see lines 110 and 118.  

Kind regards,

Xin Zhao, Guangju Wang

Reviewer 3 Report

There were many flaws and mistakes in the manuscript, the results can’t supply enough evidence for the conclusion.

Introduction

I don’t think that authors had reviewed the development of skeletal muscle growth of broiler completely, and supplied enough literatures of the progress for the environment factors on the development of skeletal muscle growth.

Authors should complete the introduction, and supply the evidence for the research, and why should determine the mentioned items.

Materials and Mehods

76-77 A total of 164 21-day-old male Arbor Acres (AA) broilers were purchased from the hatchery. 21-day-old chicken cann’t be brought form hatchery.

80-81 The ammonia concentrations of the two groups 80 were 0±3 ppm and 35±3 ppm respectively. Author had better describe how to supply ammonia and control the ammonia level.

94-98 euthanized using the cervical dislocation, and then the half weight of breast muscle and thigh muscle were recorded. Blood serum, liver tissue, and breast muscle were obtained and stored at -80℃ for further analyses. The part of the breast muscle tissue was taken, soaking in the paraformaldehyde for the paraffin sections made. The logical is confused. Firstly, anesthesia, and then drop blood, and then remove breast, and so on.

188-189 TP, ALB and BUN can be used as markers of protein metabolism (23).However, TP, ALB, BUN is used to evaluate the diet protein utilization, not for body protein metabolism. Why serum total protein (TP), albumin (ALB) and blood urea nitrogen (BUN) was determined in this study? Did it can help to explain ammonia inhibiting muscle synthesis?

204 IGF-1 can regulate the synthesis and degradation of proteins. Why didn’t determine IGF-1 in serum? mRNA expression isn’t always same with protein expression.

Actually, muscle development in grower broilers mainly based on satellite cell proliferation and hypertrophy, which affected by nutrition and environment. The F-box proteins (FBXO) are crucial for the process of ubiquitin-dependent proteolysis in eukaryotes. Environmental stressors can lead to enhanced muscle proteolysis. The F-box protein family plays a crucial role in protein ubiquitination degradation, and the protein FBXO32 is a common marker for accelerated proteolysis and atrophy processes. MYF5 is an important transcriptional activator that induces fibroblast differentiation into myoblasts, promotes transcription of muscle-specific target genes, and plays a role in muscle differentiation. MYOG is essential for muscle homeostasis and for regulating myocyte fusion by determining the number and size of muscle fibers. Why just measured the expression of MyHC-FWM, MyHC-FRM, MyHC-SM, not included F-box proteins(FBXO), MYF5, MYOG?

usually, sentence shoud use passive voice, not active voice. Many sentences are difficult to be understood. 

Author Response

Dear reviewer,

Thanks for your helpful comments and suggestions, as you said, the structure of the introduction is not systemic enough, and we have added supportive information to the introduction. 

Thank you for you pointed the problem in the experimental design, we have corrected the expression of the broiler sources, please see lines 99 and 100.

We have added the ammonia supply approach description: ammonia was delivered and monitored with precise facilities during the entire experiment, please see lines 103 - 106.   According to the previous review articles, the TP and albumin (ALB) levels in the serum reflect the state of protein absorption and metabolism, and blood urea nitrogen (BUN) is a nitrogenous end product of protein metabolism, which reflects the physiological reactions of body protein intake and metabolism, liver and kidney function.   As for why we measure the expression of IGF-1. when we consult research on rats and humans, the beneficial evidence of IGF-1 to skeletal muscle was almost to measure the mRNA expression of IGF-1. Then we measure the mRNA expression. Definitely, measuring both of them is the best approach.

Thanks for your detailed introduction. Indeed, the F-box proteins, MYF5, MYOG are crucial factors in protein metabolism. However, this part is not core of this studies. We just want to show that the growth of skeletal muscle was inhibited and the morphology changes under the influence of ammonia. Furthermore, because of the limited experimental period, we did not measure the parameters mentioned above.  

Kind regards,

Xin Zhao, Guangju Wang

Reviewer 4 Report

The paper by Zhao et al examines how exposure to atmospheric ammonia affects skeletal muscle growth in broilers. The authors found that exposure to 35 ppm of ammonia for 21 days resulted in decreased muscle mass, which was likely due to an increase in myostatin. Myostatin is a protein that inhibits the activation of signaling pathways that promote muscle growth.

The manuscript requires careful revision for English language errors and tense usage. This means that there are grammatical and tense errors that need to be fixed to improve the quality of the writing. 

There is an inconsistency between the abstract and results and discussion section regarding BUN levels. The abstract reports a decrease in BUN, while the results and discussion section reports an increase. This discrepancy must be addressed to ensure accuracy and clarity.

Table 5 footnote requires attention.

H&E slides should be provided with scale and magnification information. It is important to provide scale and magnification information to help readers interpret the images accurately.

While the manuscript discusses the negative effects of ammonia on skeletal muscle growth in broilers, there are several articles that have already established this correlation through various pathways. It would be beneficial if the paper could highlight what makes this research different or unique. This means that the authors should explain what is new or different about their research compared to previous studies on the topic.

The manuscript should also address the fact that correlation does not necessarily imply causality. This means that while the authors have found a correlation between atmospheric ammonia exposure and decreased muscle mass, they should acknowledge that there may be other factors or mechanisms involved and that further research is needed to establish causality.

The whole manuscript requires careful revision for English language errors and tense usage. 

Author Response

Dear reviewer,

Thanks for your helpful comments and suggestions, we have carefully checked mistakes and revised the manuscript.

Sorry we made a mistake in BUN content in the abstract, and we have corrected that.

We have adjusted the Table 6 position and footnote, please see line 202.

We have added the scale information in the Figure 1, please see line 205.

Thanks for your comments of the topic. Indeed, there are several articles have demostrated the relationship of muscle growth in broiler and atmosphric ammonia. But the main aim the this study is to preliminary explore the mechanism of ammonia effect on skeletal muscle growth. We believe that, so far, there is little study on that in the field of broilers. As for the expression of the conclusion, we shouldn't conclude the mechanism in one sentence, because our data is not enough to prove the exist of this mechanism, it can be the correlation between factors and results. Therefore, we adjusted the conclusion.

Kind regards,

Xin Zhao, Guangju Wang

Round 2

Reviewer 3 Report

Authors didn't marked the revision sentence, or use the modification model. So, I can't revise the revised manuscript. there are many grammer mistakes in the manuscript, which should be revised by native English writing.

Authors didn't marked the revision sentence, or use the modification model. So, I can't revise the revised manuscript. there are many grammer mistakes in the manuscript, which should be revised by native English writing.

Author Response

Dear reviewer,

I am sorry for the inconvenient on review manuscript. Indeed, we used a modification mode to show where we revised in manuscript, this time we highlight the revised part in yellow. I wish you can review in convenient way.

Kind regards,

Xin Zhao, Guangju Wang

Round 3

Reviewer 3 Report

The manuscript should be ask native Engish writing to polish again. There are many English grammer mistakes in the manuscript. 

There are many English language problem in sentences,  included voive and tense. it should be positive voice, not active voice.  Past tenses shoulu be used .

24-27 the ammonia group birds had a lower breast muscle, thigh muscle, and dressed weight, the blood urea nitrogen content increased, skeletal muscle fibre diameter shortened, the expression of myostatin increased, and the expression of myosin heavy chain-FWM and myosin heavy chain-FRM decreased significantly.

65-66 a previous study showed that fetal mice with MSTN knockout showed a larger proportion of fast muscle and glycolytic phenotype (17)

80-81 A total of 164 0-day-old male Arbor Acres (AA) broilers were purchased from the hatchery and started the test after 21 days of feeding

103-105 At the end of each test period (7d and 21d), birds of the control group and the ammonia group (one bird each replicate) were euthanized using the cervical dislocation after anesthesia, collected blood sample and remove the half breast muscle and thigh muscle

261-263 In conclusion, after 21 days of ammonia exposure, the expression of MSTN was increased, resulting in a reduced muscle protein synthesis rate, increased degradation rate, and tapering in the diameter of the muscle fibres.

Author Response

Dear reviewer,

Thanks for your suggestions about the English language problems. In fact, we tried our best to revise the English grammar and the structure of sentences. However, there might still be many problems in the manuscript. Therefore, I would like to apply for the English retouching service at MDPI.

We made adjustments to the sentences you mentioned and in highlight. Please review that to see if it is correct. 

As for the words of the manuscript, we tried our best to extend the manuscript. Since our research is a relatively new perspective on the ammonia effect on broilers, we can cite a few studies, but not that many.  We want to focus on the effects of ammonia on muscle growth. Thus, we wish our manuscript could be accepted by "ANIMALS".

Kind regards,

Xin Zhao, Guangju Wang